# Improving the Photocatalytic Activity of Ti_3_C_2_ MXene by Surface Modification of N Doped

**DOI:** 10.3390/ma16072836

**Published:** 2023-04-02

**Authors:** Lidan Cui, Jianfeng Wen, Quanhao Deng, Xin Du, Tao Tang, Ming Li, Jianrong Xiao, Li Jiang, Guanghui Hu, Xueli Cao, Yi Yao

**Affiliations:** 1Key Laboratory of Low-Imensional Structural Physics and Application, Education Department of Guangxi Zhuang Autonomous Region, College of Science, Guilin University of Technology, Guilin 541000, China; cld930429@163.com (L.C.);; 2Guangxi Collaborative Innovation Center for Water Pollution Control and Water Safety in Karst Areas, College of Environmental Science and Engineering, Guilin University of Technology, 319 Yanshan Street, Guilin 541000, China

**Keywords:** photo-catalytic, solvent heat treatment, N-Ti_3_C_2_ MXene, Methyl Orange Removal

## Abstract

Methyl orange dye (MO) is one of the azo dyes, which is not only difficult to degrade but also hazardous to human health, therefore, it is necessary to develop an efficient photocatalyst to degrade MO. In this paper, a facile and low-cost elemental doping method was used for the surface modification of Ti_3_C_2_ MXene, i.e., nitrogen-doped titanium carbide was used as the nitrogen source, and the strategy of combining solvent heat treatment with non-in situ nitrogen doping was used to prepare N-Ti_3_C_2_ MXene two-dimensional nanomaterials with high catalytic activity. It was found that the catalytic efficiency of N-Ti_3_C_2_ MXene materials was enhanced and improved compared to the non-doped Ti_3_C_2_ MXene. In particular, N-Ti_3_C_2_ 1:8 MXene showed the best photo-catalytic ability, as demonstrated by the fact that the N-Ti_3_C_2_ 1:8 MXene material successfully degraded 98.73% of MO (20 mg/L) under UV lamp irradiation for 20 min, and its catalytic efficiency was about ten times that of Ti_3_C_2_ MXene, and the N-Ti_3_C_2_ photo-catalyst still showed good stability after four cycles. This work shows a simplified method for solvent heat-treating non-in situ nitrogen-doped Ti_3_C_2_ MXene, and also elaborates on the photo-catalytic mechanism of N-Ti_3_C_2_ MXene, showing that the high photo-catalytic effect of N-Ti_3_C_2_ MXene is due to the synergistic effect of its efficient charge transfer and surface-rich moieties. Therefore, N-Ti_3_C_2_ MXene has a good prospect as a photo-catalyst in the photocatalytic degradation of organic pollutants.

## 1. Introduction

With the progressive increase in population and the rapid expansion of business over the past decade, various types of domestic sewage and industrial wastewater have rapidly increased, making the water pollution problem more serious and causing serious damage to the ecological environment [1,2,3]. In particular, azo dyes are discharged from wastewater due to their anti-biodegradability. Excessive amounts of azo dyes are teratogenic, mutagenic, and carcinogenic, posing a highly toxic hazard to plants, animals, and humans [4,5] Therefore, there is an urgent need for efficient and cost-effective water treatment technologies. So far, many methods have been developed, such as adsorption, redox photochemical degradation, membrane filtration, and photocatalysis [6,7] have been widely used to remove pollutants from water. Among them, photocatalytic technology is gaining popularity owing to its several benefits, which include environmental friendliness, high efficiency, stability, and a high degradation rate [8,9].

MXenes [10], a novel two-dimensional substance, has several applications in photo-catalysis [11] and high electrical conductivity owing to its enormous specific surface area [12], many active sites [13], and the presence of numerous functional groups (-O, -OH, -F, etc.) with good hydrophobicity and controllable interlayer distances [14,15,16,17]. MXenes follow the general formula of M_n+1_X_n_T_x_, where n+1 (n = 1, 2, 3) is the early transition metal layer, and M is intercalated with n layers of carbon or nitrogen X. Since the first report of MXene in 2011 [18], more than 30 MXene such as Ti_3_C_2_, Ti_3_CN, TiNbC, V_2_C, Mo_2_C, Nb_2_C, and Y_2_CF_2_ have been successfully prepared [19,20,21]. The general MXene is obtained by the selective removal of atoms from the MAX phase by treatment with hydrofluoric acid or other fluorine sources. The precursors of MXene MAX phase are a collective term for a ternary layered ceramic material, where M is a pre-transition metal (e.g., Ti, Sr, V, Nb, Cr, Ta, Zr, Mo, etc.), A is the main group II or IV element, and X is C or N, etc. [22,23,24,25,26,27]. Ti_3_C_2_, a member of the MXenes family, has superior electrical conductivity, a large specific surface area, and many active sites, which may result in the separation and transmission of photoexcited electron-hole pairs. At the same time, it is simple to create composite materials using Ti_3_C_2_ and other photo-catalysts because of their excellent flexibility and distinctive layered structure. In the case of Ti_3_C_2_ composites, Ti_3_C_2_ not only increases the absorption of light by its blackbody but also causes rapid carrier migration and inhibits its compounding [28]. Therefore, Ti_3_C_2_ is often utilized as an alternative noble metal [29] co-catalyst to boost the complexes’ photo-catalytic activity [30,31,32,33]. We note that it has been reported in the literature that Ti_3_C_2_ with -F and -OH end groups are a narrow band gap semiconductor [34], and considering that Ti_3_C_2_ exfoliated from HF acid liquid phase will have -F and -OH end groups and may also have photo-catalytic properties. Therefore, we investigated the photocatalytic activity of Ti_3_C_2_. Unfortunately, the photosynthesis performance of Ti_3_C_2_ itself was not satisfactory.

Considering that Ti_3_C_2_ can be improved by changing the elemental composition and adjusting the surface functional groups to improve the performance of two-dimensional accordion-like Ti_3_C_2_ nanomaterials, we used the surface modification of Ti_3_C_2_ utilizing introducing heteroatoms to investigate its photocatalytic properties. It has been discovered that ecologically friendly urea (CH_4_N_2_O) in high-concentration nitrogen alcohol is an effective source of liquid nitrogen doping and that the liquid N doping source can retain excellent contact with Ti_3_C_2_ to achieve high N doping without stacking N-Ti_3_C_2_ nanospheres [35,36]. The modulation of nitrogen-doped Ti_3_C_2_ nanomaterials is greatly enhanced in terms of carrier density, surface energy, and surface reactivity.

Currently, there are many research reports on N-Ti_3_C_2_ in the fields of batteries [37], capacitors [38], and electrocatalysis [39], however, there are fewer reports on the application of N-Ti_3_C_2_ in the field of photocatalysis. Thus, in this paper, urea-saturated alcohol solution as a liquid nitrogen source, and a simple and controllable strategy combining solvent thermal treatment and non-in situ nitrogen doping were used to prepare N-Ti_3_C_2_ MXene 2D nanomaterials with high catalytic activity. Transforming the usual way of N-Ti_3_C_2_ nanomaterials, we applied it as the main catalyst for the first time in the field of photocatalysis and confirmed that it could significantly improve the photocatalytic degradation of MO. In addition, the morphological and physicochemical properties of the materials were thoroughly examined, and the reaction mechanism of the photo-catalysts was discussed through experimental and theoretical studies. 

## 2. Materials and Methods

### 2.1. Preparation of N-Ti_3_C_2_ Catalyst

Ti_3_AlC_2_ (99.7%), urea (AR), dimethyl sulfoxide (DMSO, AR), hydrofluoric acid (HF, AR), and anhydrous ethanol (99.5%) were purchased from Guilin Bell Experimental Equipment Co. All chemical reagents were analytical grade without additional purification, and all experiments utilized clean water.

The synthetic roadmap of the N-Ti_3_C_2_ catalyst is shown in Figure 1, and its structure is schematically shown in Figure 2.

Synthesis of Ti_3_C_2_: Ti_3_C_2_ was synthesized by etching off the aluminum element in Ti_3_AlC_2_. Firstly, 2 g Ti_3_AlC_2_ powder was mixed with 20 mL of 30% hydrofluoric acid at room temperature for 5 h. Secondly, the etched material was centrifuged by centrifuge to collect the black powder and washed with deionized water to a pH of about 6–7, and placed in a dryer for 12 h at 60 °C.

The synthesis route of N-doped Ti_3_C_2_: Firstly, 0.5 g of the above-dried Ti_3_C_2_ was placed in 15 mL of dimethyl sulfoxide (DMSO) and agitated for 24 h for intercalation. The powder was recovered by centrifugation, washed several times with deionized water, and then dried for 12 h at 60 °C in a drier. Next, take the intercalated Ti_3_C_2_ and urea in the ratio of 8:1 put it into 60 mL anhydrous ethanol for mixing, put it into the ultrasonic machine for 30 min to mix well, and transfer it to the stainless-steel autoclave lined with polytetrafluoroethylene for solvent heat reaction at 180 °C for 12 h. Finally, the material in the reaction kettle was collected by centrifugation, washed with deionized water and anhydrous ethanol until the pH was about neutral, and put into a dryer for 12 h at 60 °C to obtain N-doped Ti_3_C_2_ powder. 

### 2.2. Materials Characterization

By using energy-dispersive X-ray spectroscopy (EDS) detectors and a field emission scanning electron microscope (FESEM, SU5000, Hitachi, Tokyo, Japan), researchers were able to examine the morphological characteristics and elemental composition of the samples. X-ray photoelectron spectroscopy (XPS, ESCALAB-250XI) analyzed the valence and chemical composition of the elements. X-ray diffraction spectroscopy (XRD, Bruker D8 Advanced) measured the crystal structure of the samples. Fourier infrared spectroscopy (FTIR) was used to measure the functional groups and chemical bonds on the surface of the samples. Optical properties were tested by UV-Vis spectrophotometer (DRS, Shimadzu Lambda 750, Tokyo, Japan). 

### 2.3. Electrochemical Measurements

The prepared sample coated on FTO glass served as the working electrode, platinum wire served as the counter electrode, the Hg/Hg_2_Cl_2_/KCl (saturated) electrode served as a reference, and 0.5 mol/L aqueous solution served as the electrolyte, photochemical measurements were carried out using an electrochemical workstation (CHI860B). The produced photocurrent was measured while a solar simulator was irradiated with a 10-s lamp on/off cycle. The irradiation intensity was 1000 W/m^2^ and the effective area of the sample was 1 cm^2^.

### 2.4. Photocatalytic Performance

The photocatalytic performance of N-Ti_3_C_2_ was tested by methyl orange dye MO degradation. The sample numbers obtained at different scales will be used in this experiment, and the light source used is a 1000 W mercury lamp (Wavelength range 320 nm–390 nm, main peak value 365 nm) to emit ultraviolet light. Take 50mL of methyl orange dye with a concentration of 20 mg/L mixed with 50 mg N-Ti_3_C_2_ and pour it into the test tube, sonicate for some time to make it mix evenly, and then put it into a photochemical reactor (BL-GHX-1D) and stir under dark conditions for 30 min to achieve adsorption resolution equilibrium, turn on the light source and take 3–4 mL every 10 min. The resulting solutions were detected with a UV-Vis spectrophotometer for absorbance at 464 nm (characteristic wavelength of MO) to determine the degree of degradation.

## 3. Results and Discussion

### 3.1. SEM and EDS Analysis

In the chemical etching of aluminum in Ti_3_AlC_2_ as shown in Figure 3a,b, SEM can observe that Ti_3_C_2_ MXene shows a unique accordion-like multilayer structure, revealing the effective production of Ti_3_C_2_ through HF etching of Ti_3_AlC_2_. It is well known that at a certain temperature and high pressure, due to the low boiling point and high mobility of ethanol, it may easily lead to the diffusion of urea molecules into the interlayer space during the solvent heat treatment. The morphology of the nitrogen-doped Ti_3_C_2_ MXene is shown in Figure 3c,d was also well preserved after the solvent thermal reaction, in which the spacing between some of the layers became larger, the surface of the sample was rougher and more complex, and the layer surface was attached with uniform and fine nanoparticles. To further elucidate the structure of N-Ti_3_C_2_ MXene nanomaterial, EDS elemental mapping of individual elements in N-Ti_3_C_2_ MXene nanomaterial was performed (Figure 3e), and X-ray spectra (EDS) showed that Ti, C, N, F, and O elements were uniformly distributed in the prepared samples. These characterizations strongly demonstrate the successful preparation of N-Ti_3_C_2_ MXene 2D nanomaterial. The formation of N-Ti_3_C_2_ expands the interlayer distance, resulting in an increased specific surface area and more active sites, which is more favorable for carrier transport and separation.

### 3.2. XRD and FTIR Analysis

The X-ray powder diffraction was used to examine the crystalline and phase structure of the material (XRD). As shown in Figure 4a, the typical four prominent peaks of Ti_3_AlC_2_ (JCPDS No. 52-0875) are distributed at 9.517°, 19.154°, 39.037°, and 60.257°, corresponding to the four crystallographic planes (002), (004), (104), and (110), respectively. The strongest XRD diffraction peak of Ti_3_AlC_2_ after HF etching (104) crystalline plane disappears and (002) and (004) crystalline planes are shifted to lower angles compared to Ti_3_C_2_. The results show that the layered Ti_3_C_2_ MXene was successfully prepared. with the addition of the nitrogen dopant, the diffraction peaks of the (002) crystalline plane corresponding to N-Ti_3_C_2_ 1:2, N-Ti_3_C_2_ 1:5, N-Ti_3_C_2_ 1:8 and N-Ti_3_C_2_ 1:10 shifted from 9.517° to 6.94°, 6.8°, 6.79° and 6.75° relative to Ti_3_C_2_ is particularly obvious. This indicates that the interlayer distance of N-Ti_3_C_2_ MXene increases with the doping of nitrogen, which gives it a larger specific surface area and exposes more active sites. In addition, we found diffraction peaks (101) and (211) of titanium dioxide based on XRD diffraction patterns, indicating that a small portion of titanium dioxide was also formed during the nitrogen doping process [40]. The surface functional groups and bonds of the sample were measured by Fourier infrared spectroscopy (FTIR), as shown in Figure 4b, and some surface -F groups of Ti_3_C_2_ were substituted by hydroxyl groups through anhydrous ethanol due to the abundance of hydroxyl groups in anhydrous ethanol, i.e., the sharp characteristic peaks near 3606 and 3738 cm^−1^ represent free -OH. Since the N atom forms polar covalent bonds with the H atom and is easily adsorbed to the Ti_3_C_2_ surface, the characteristic peaks near 3130 and 3109 cm^−1^ are the N-H vibrational peaks of the bonding, the vibrational peaks at 1350–1550 cm^−1^ can be attributed to the vibrational stretching of -CH_3_ and the C-N stretching vibrational peaks in the urea molecule, and the characteristic peaks near 580 cm^−1^ corresponds to the stretching vibration of Ti-C. The presence of these functional groups can promote electron transfer and enhance electrical conductivity, while the doping of N atoms can effectively improve the electron transport network and ion transport channels of Ti_3_C_2_ MXene materials [41].

### 3.3. XPS Analysis

X-ray photoelectron spectroscopy was used to investigate the chemical makeup and surface state of N-Ti_3_C_2_ MXene (XPS). As illustrated in Figure 5a. The obvious peaks of Ti, O, C, and F elements could be seen by the full scan spectrum, compared to the XPS curve of N- Ti_3_C_2_ material with an extra minor peak of N element. In addition, N-Ti_3_C_2_ materials have higher peak O element intensities than Ti_3_C_2_ materials, and the atomic concentration of O increases from 18.33% to 29.15%, indicating that part of Ti_3_C_2_ is converted to TiO_2_ in the hydrothermal process. Among them, it should be noted that the atomic concentration of F is 11.9%, indicating the presence of a considerable amount of F termination groups on the surface of N-Ti_3_C_2_ MXene. Figure 5b represents the original Ti_3_C_2_ MXene with peak I at 685.0 eV representing the C-Ti-F functional group and peak II at 686.5 eV representing AlFx [42], with a shift in the peak position after nitrogen doping, probably due to a change in the elemental valence state and the appearance of a new diffraction peak, i.e., peak III at 689.18 eV, representing F-C [43]. Similarly, peak I at 529.9 eV in the O 1s region of the spectrum in Figure 5c represents the C-Ti-O functional group, peak II at 531.1 eV represents the C-O functional group, and peak III at 533.0 eV represents the C-Ti-OH functional group [44]. As shown in Figure 5d, peaks I at 455.1 eV and 461.1 eV in the Ti 2p region of the XPS spectrum of Ti_3_C_2_ MXene refer to the C-Ti functional group, whereas the second expanded peak II at 456.6 eV alludes to the C-Ti-F functional group [42]. In addition, the XPS spectrum of Ti 2p in N-Ti_3_C_2_ MXene has a new pair of peaks III (Ti 3p/2 and Ti p/2) appearing at 459.2 eV and 464.88 eV, respectively, inferred to be possibly from the oxidation product TiO_2_ formed during the hydrothermal treatment [45], which corresponds to the (101) and (211) diffraction peaks found in the XRD diffraction energy spectrum corresponding to the formation of TiO_2_, both confirming the formation of TiO_2_. From Figure 5e, it can be seen that the XPS spectra of C 1s found in the pristine Ti_3_C_2_ MXene have three diffraction peaks with different valence states, which may originate from external functional groups; peak I is 282.0 eV, representing the C-Ti-(O/OH/F) functional group; peak II is 284.8 eV, representing the C-C functional group; and peak III is 286.5 eV, representing the C-Hx/C-O functional group. One more diffraction peak can be seen in the XPS spectrum of C 1s in N-Ti_3_C_2_ compared to Ti_3_C_2_ after doping with nitrogen, i.e., peak IV is 289.1 eV, representing the C-N functional group (visible in the Annex) [45].

Figure 5f shows the XPS spectra of N 1s. Based on the high-resolution spectra of N 1 s, it can be shown that nitrogen is successfully doped into Ti_3_C_2_ and three peaks are identified near 396.58 eV, 400.18 eV, and 401.98 eV, which correspond to three nitrogen functional groups [15,35,46]: (i) nitride, abbreviated as N-Ti (396.58 eV), (ii) pyrrole nitrogen, abbreviated as N-5 (400.18 eV) and (iii) quaternary nitrogen, abbreviated as N-Q (401.98 eV). The peak at 396.58 eV originates from the substitution of nitrogen for carbon atoms to form mainly N-Ti (nitrides), which indicates that the N atom reacts not only with the C atom, but also bonds with the Ti atom. In contrast, the two peaks located near 400.18 eV and 401.98 eV originate from the -N functional group and surface adsorption, respectively [35]. It can be inferred that nitrogen is mainly present in the form of surface adsorption and pyrrole nitrogen (N-5), and the new peak found near 289.1 eV in the carbon elemental analysis spectrum also indicates the formation of the -N functional group. 

### 3.4. Transient Photocurrent Response

Today, photocurrent is widely considered to be the most effective evidence for charge separation in heterogeneous structured photo-catalysts [42,47,48]. Typically, the value of photocurrent indirectly reflects the ability to generate and transfer photo-excited charge carriers under irradiation, which correlates with photo-catalytic activity. 

Using the use of transient photocurrent response, the migration and separation traits of the photo-generated electron-hole pairs of the samples were examined. The transient photocurrent curves of N-Ti_3_C_2_ are shown in Figure 6. Among all the samples, the photocurrent of Ti_3_C_2_ was the highest, while the photocurrent of N-Ti_3_C_2_ 1:8 was the lowest, and the photocurrent of Ti_3_C_2_ was much larger. The photocurrent magnitudes of Ti_3_C_2_ and N-Ti_3_C_2_ 1:8 are not consistent with their photo-catalytic activities. It may occur as a result of the possibility that oxygen molecules adsorbed on the surface of N-Ti_3_C_2_ can interact with unbound electrons to form negatively charged O_2_^−^.

It is known that the current is proportional to the mobility (μ) and density (n) of the charge carriers [49].
I = qnμFA
where q is the electron charge, F is the electric field, and A is the cross-sectional area. F and A were the same across all samples included in the investigation. This shows the great difference in material carrier mobility and that high photocurrent does not imply an abundant carrier density [50].

The transient photocurrent density of different samples was measured with three electrodes, and the order of the magnitude of photocurrent was Ti_3_C_2_ > N-Ti_3_C_2_ 1:2 > N-Ti_3_C_2_ 1:10 > N-Ti_3_C_2_ 1:5 > N-Ti_3_C_2_ 1:8, which is the opposite order of photo-catalytic activity, so it can be seen that there is no absolute relationship between the magnitude of photocurrent and photocatalytic activity.

### 3.5. Photocatalytic Degradation Activity and Cycling Experiments of MO

The photocatalytic ability of nitrogen-doped Ti_3_C_2_ MXene two-dimensional material was investigated by photocatalytic degradation of MO. The degradation efficiency was defined as *η* = (C_0_ − C_t_)/C_0_, where C_0_ is the initial concentration of methyl orange and C_t_ is the concentration of methyl orange at time t. Blank experiments with degradant solutions under the same conditions without a catalyst were also performed for comparison. In Figure 7a, Ti_3_C_2_ MXene materials with nitrogen doping ratios of 1:2, 1:5, 1:8, and 1:10 were tested. Compared to pure Ti_3_C_2_ MXene, N-Ti_3_C_2_ 2D nanomaterials with high surface area and large pore volume due to N-Ti_3_C_2_ 1:2, N-Ti_3_C_2_ 1:5, N-Ti_3_C_2_ 1:8 and N Ti_3_C_2_ 1:10 two-dimensional nanomaterials all exhibited better photo-catalytic ability. Especially, N-Ti_3_C_2_ 1:8 had the best photo-catalytic ability to degrade MO under UV light, and it degraded 98.73% of the methyl orange dye after 20 min of irradiation, and the removal rate of MO was 99.93% at 40 min. Figure 7b shows the pseudo-first-order kinetic plot of MO degradation with the equation: −ln(C_t_/C_0_) = kt. Figure 7c depicts the kinetic rate constant (k) estimated using the pseudo-first-order kinetic model, as well as the variation of k for all samples visually, N-Ti_3_C_2_ 1:2, N-Ti_3_C_2_ 1:5, N-Ti_3_C_2_ 1:8 and N-Ti_3_C_2_ 1:10 for The kinetic rate constants for MO removal by 2D nanomaterials were 0.00574, 0.02387, 0.14669, 0.18717, and 0.02959 min^−1^. The above results show that the N-Ti_3_C_2_ 1:8 material shows the best photocatalytic performance with abundant active sites. The main reason may be that the nitrogen-doped materials have more abundant surface functional groups, which improve the charge transfer efficiency and enhance the electrical conductivity of the materials.

In addition, the reusability of N-Ti_3_C_2_ 1:8 2D nanomaterial was carried out to assess the stability. As seen in Figure 7d, the catalytic activity of the N-Ti_3_C_2_ 1:8 2D nanomaterial for the degradation of methyl orange dye still reached 94.87% after four cycles. The catalytic degradation efficiency decreased only slightly after 4 cycles, and the slight decrease might be due to the loss of photocatalyst during the recovery process. The results demonstrate that the N-Ti_3_C_2_ 1:8 two-dimensional nanomaterials have good chemical stability. 

### 3.6. Effect of Dye Concentration on Degradation Rate 

The experimental dye concentration gradient ranged from 20–60 mg/L, and it can be seen from Figure 8 that the photocatalytic degradation rate decreases sharply with the increase in dye concentration. There are two reasons for this: Firstly, at a certain amount of catalyst, the greater the concentration of dye, the slower the decrease in chromaticity, and the longer the time required for complete degradation [51]. Secondly, the solution light transmittance decreases with increasing dye concentration, the radiant energy absorbed by the catalyst decreases, and the photocatalytic efficiency becomes poor [52,53]. In addition, we also compared different two-dimensional material-based composite photocatalysts listed in Table 1.

### 3.7. Optical Properties

The UV-vis absorption properties and band gap of the materials were investigated by UV-vis diffuse reflectance spectroscopy. From Figure 9a, it can be seen that the absorption range of N-Ti_3_C_2_ is wider, and the absorption edge extends to 800 nm. The absorption of N-Ti_3_C_2_ is higher than that of Ti_3_C_2_, which is related to the doping of nitrogen elements. Moreover, a new absorption band appears at the position of the peak around 400 nm after nitrogen doping. Considering that the previous XRD and XPS tests showed the appearance of TiO_2_ in the sample after nitrogen doping, this absorption should be caused by TiO_2_. It can be seen from Figure 9b that nitrogen doping reduces the band gap, and the band gap of the material after N doping is reduced from 1.376 eV to 1.145 eV.

### 3.8. Proposed Degradation Pathway and Photocatalytic Mechanism

In Figure 10, to further explore the photo-catalytic active substance, is to conduct a capture experiment of the active substance. In the degradation of MO by N-Ti_3_C_2_ 1:8 2D nanomaterial, ethylenediaminetetraacetic acid disodium salt (EDTA-2Na) as a hole scavenger (h^+^), isopropyl alcohol (IPA) as a hydroxyl radical (•OH) scavenger, sodium iodate (NaIO_3_) as an electron (e^−^) scavenger and p-benzoquinone (BQ) as superoxide radical (•O_2_^−^) scavengers were analyzed for the active substances that play a major role in the photocatalytic degradation of methyl orange dye. The catalytic activity of N-Ti_3_C_2_ 1:8 2D nanomaterial for MO degradation was found to be unaffected by the presence of EDTA-2Na, showing that the photo-catalytic active component was not h^+^. And the catalytic activity decreased significantly after the addition of IPA, NaIO_3_, and BQ, which proved that •OH, e^−^, and •O_2_^−^ were the photo-catalytic active substances that played the main role in the UV degradation of MO experiments.

Organic pollutants first diffuse from the contaminated solution to the surface of the photocatalyst and then desorb to the outer surface of the photocatalyst through effective adsorption and products of redox reactions. Based on the analysis of active substances, a speculative method to explain the photocatalytic activity of N-Ti_3_C_2_ material for the degradation of pollutants is proposed. As shown in Figure 11, Ti_3_C_2_ is recognized as a material with excellent electrical conductivity, which can effectively separate electrons and holes and inhibit their complexation, and N doping further enhances this property. Photo-generated electrons and holes are generated on the surface of N-Ti_3_C_2_ MXene under the photoexcitation of UV light. And some of the photogenerated electrons on the conduction band of N-Ti_3_C_2_ MXene after photoexcitation are directly involved in the degradation of MO as a major active substance. Another part of photogenerated electrons reacts with O_2_ to produce the active substance superoxide radical (•O_2_^−^) to participate in the degradation process of MO. The other part of photogenerated electrons reacts with O_2_ to produce the reactive substance superoxide radical (•O_2_^−^) to participate in the degradation process of MO [60]. The photogenerated holes left in the valence band react with adsorbed hydroxide ions (OH^−^) and H_2_O to produce hydroxyl radicals (•OH), allowing the oxidation of organic pollutants by •OH [61,62]. These highly reactive species (•O_2_^−^, •OH, and e^−^) enable rapid photodegradation of MO under UV light irradiation. This is in agreement with our active species capture experiment Figure 10.

## 4. Conclusions, Ongoing Challenges, and Perspective

In this work, N-Ti_3_C_2_ MXene two-dimensional nanomaterials were prepared by a facile and controllable strategy combining solvent heat treatment and non-in situ nitrogen doping using urea-saturated alcohol solution as a liquid nitrogen source. Its methyl orange dye removal rate was 98.73% after 20 min of UV lamp irradiation, and it still had good photo-catalytic activity after four cycles. N-doped Ti_3_C_2_ MXene would not only provide an effective electron transfer pathway but also increase the interlayer distance of MXene and expose more active sites, thus showing extraordinary photo-catalytic performance in pollutant degradation. Moreover, it further demonstrates that the electrical conductivity of N-Ti_3_C_2_ MXene can effectively transfer photo-generated electrons, resulting in a large amount of photo-catalytically active substances (•OH, e^−^, and •O_2_^−^). Therefore, N-Ti_3_C_2_ MXene two-dimensional nanomaterial can be used as an effective and promising photo-catalyst for wastewater purification.

For the present, there are some limitations to this work. The excitation light source used during the experiments is UV light, which is more energetic, and the organic pollutants degraded in the photo-catalytic study are relatively single. The cycling experiments have not yet reached efficiency saturation. There is a need to further explore the photocatalytic activity of N-Ti_3_C_2_ MXene under sunlight or visual light to degrade a variety of organic pollutants, such as antibiotics. And the photo-catalytic cycle of the material needs to be further investigated until the cycling efficiency is saturated. In addition, a toxicity assessment of the treated solution is needed to ensure that the discharged wastewater will reduce the adverse environmental impact.

## Figures and Tables

**Figure 1 materials-16-02836-f001:**
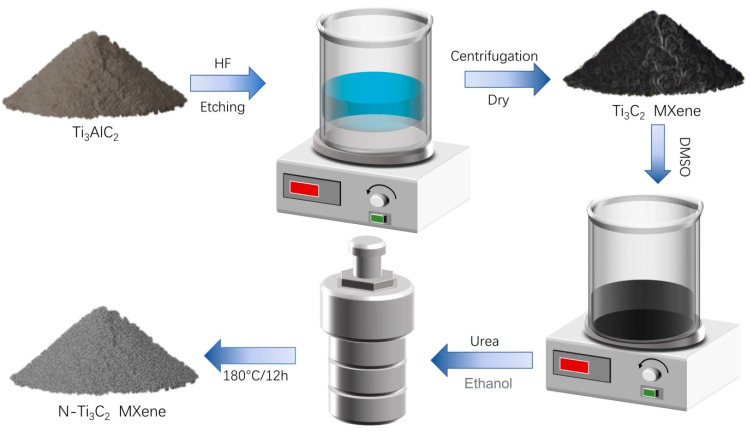
Roadmap for the synthesis of N−Ti_3_C_2_ catalyst.

**Figure 2 materials-16-02836-f002:**
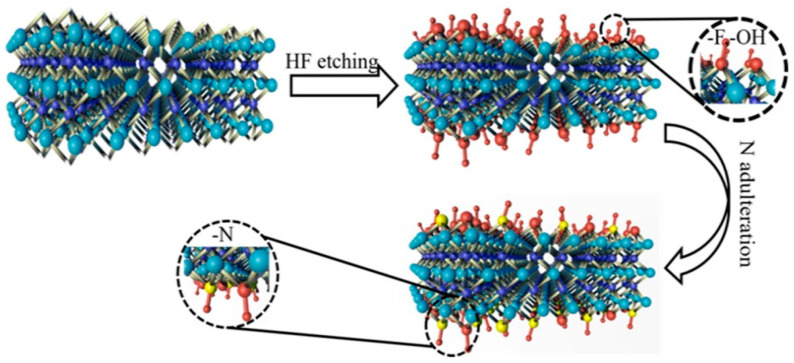
Schematic diagram of the synthetic structure of N−Ti_3_C_2_ catalyst.

**Figure 3 materials-16-02836-f003:**
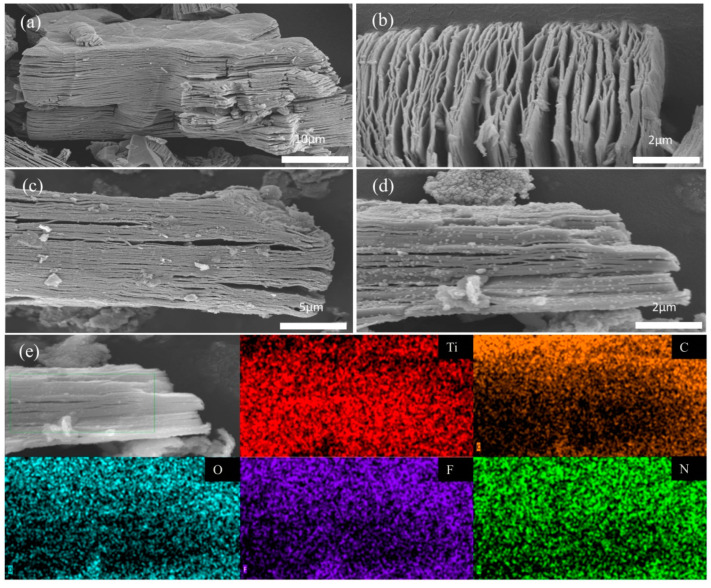
(**a**,**b**) Local magnifications of layered Ti_3_C_2_ MXene and Ti_3_C_2_; (**c**,**d**) SEM images of N−Ti_3_C_2_ MXene nanomaterials at different magnifications; (**e**) EDS elemental mapping of Ti, C, O, F and N nanomaterials of N−Ti_3_C_2_ MXene.

**Figure 4 materials-16-02836-f004:**
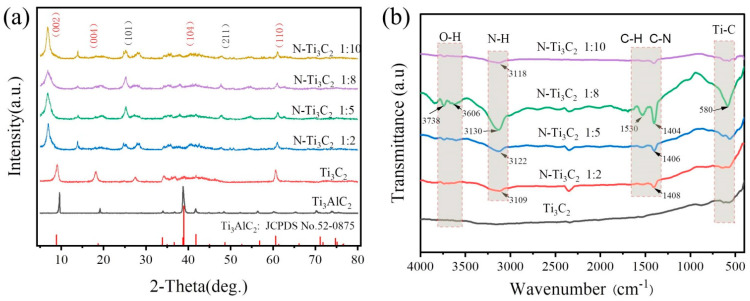
(**a**) XRD patterns of Ti_3_C_2_ MXene and N−Ti_3_C_2_ MXene nanomaterials; (**b**) FTIR spectra of Ti_3_C_2_ and N−Ti_3_C_2_ MXene.

**Figure 5 materials-16-02836-f005:**
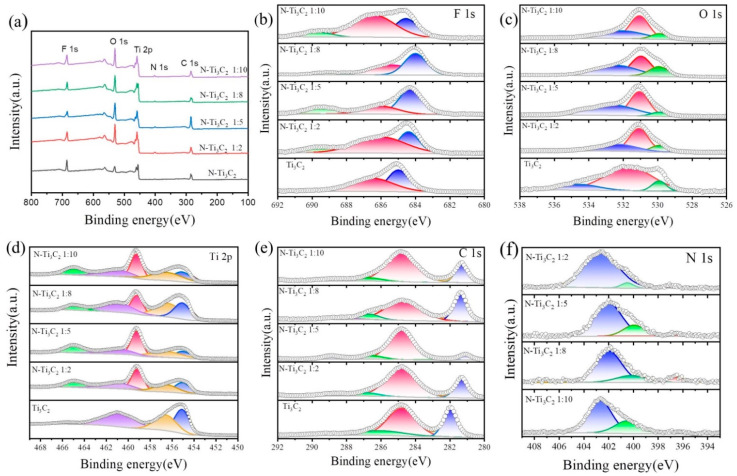
XPS spectra of the samples: (**a**) measurement scan, (**b**) F 1s, (**c**) O 1s, (**d**) Ti 2p, (**e**) C 1s, (**f**) N 1 s.

**Figure 6 materials-16-02836-f006:**
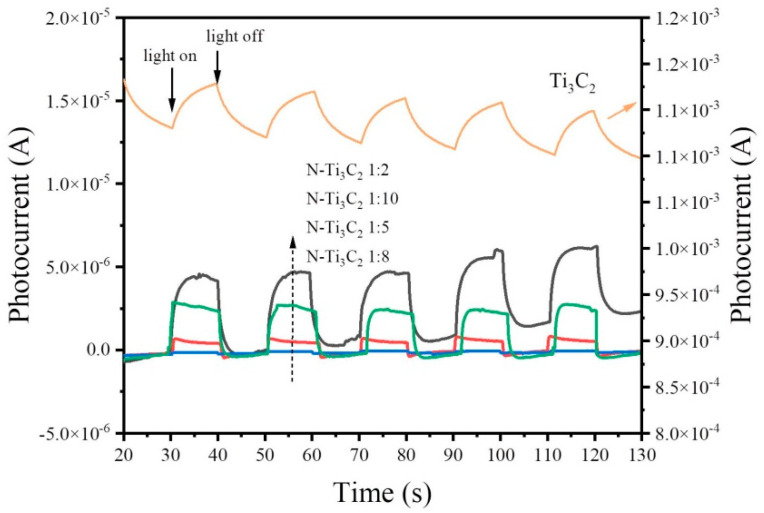
Transient photocurrents of Ti_3_C_2_ and N−Ti_3_C_2_ 1:8.

**Figure 7 materials-16-02836-f007:**
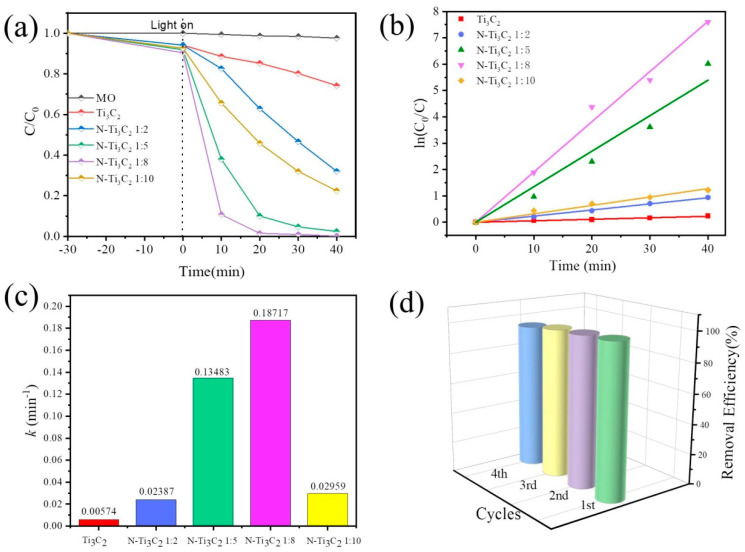
Photo-catalytic activity on (**a**) MO degradation activity, (**b**) pseudo-level diagram of MO degradation, (**c**) kinetic constants of MO degradation, (**d**) reusability of N−Ti_3_C_2_ 1:8 for MO degradation.

**Figure 8 materials-16-02836-f008:**
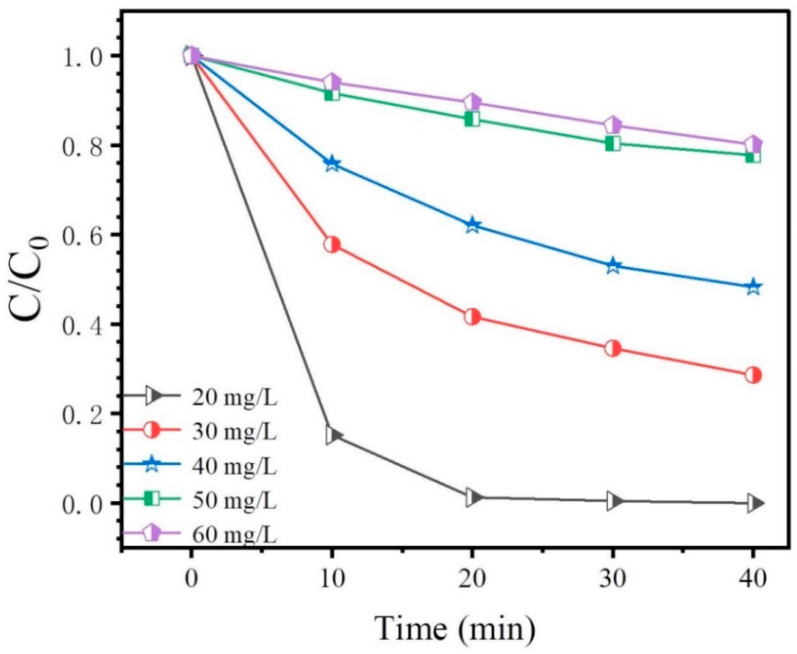
Effect of dye concentration on degradation rate.

**Figure 9 materials-16-02836-f009:**
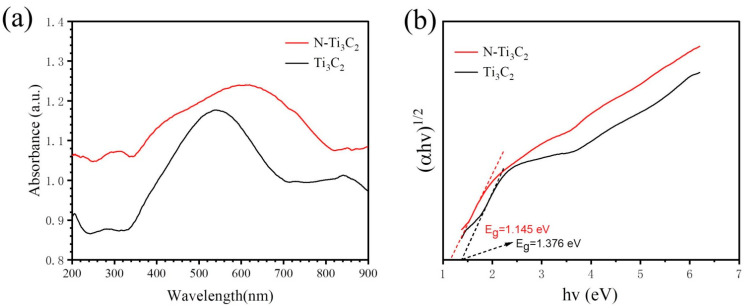
(**a**) Absorption spectra of Ti_3_C_2_ and N−Ti_3_C_2_ samples; (**b**) plots of (*α*h*ν*)^1/2^ versus photon energy (h*ν*) for band-gap energy.

**Figure 10 materials-16-02836-f010:**
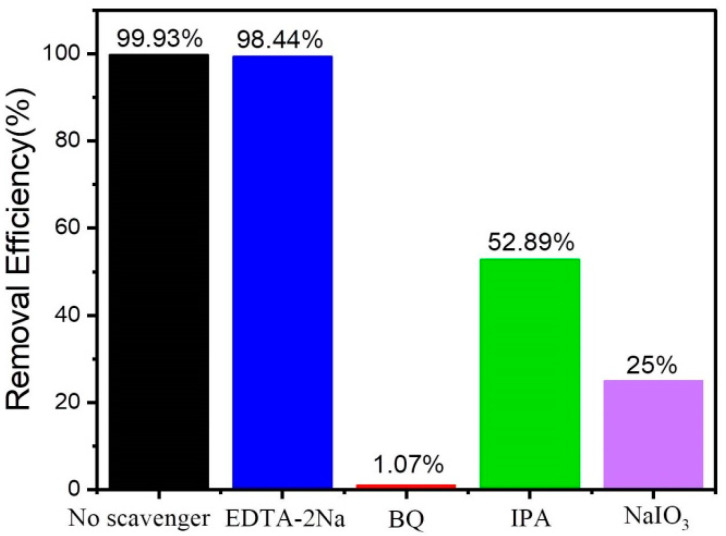
Shows the active substance capture experiment for the photo-degradation of MO by N-Ti_3_C_2_ 1:8 nanomaterial.

**Figure 11 materials-16-02836-f011:**
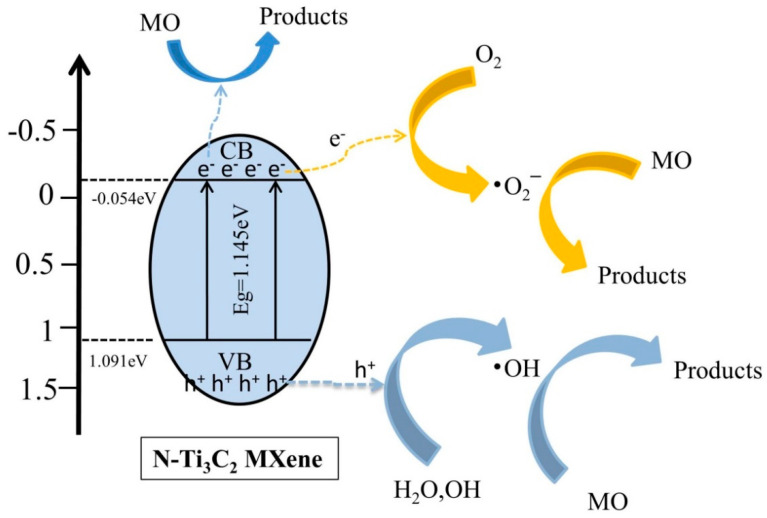
Photocatalytic degradation mechanism of N-Ti_3_C_2_ MXene.

**Table 1 materials-16-02836-t001:** Represents the photo-catalytic activity of 2D material-based composite catalysts.

Composite	Light	Composite Mass	Dye	Time (min)	Degradation Rate	Ref.
(001)TiO_2_/Ti_3_C_2_ (In-situ synthesis)	UV	10 mg	200 mL 20 mg/L Methyl orange (MO)	50	97.4%	[54]
TiO_2_/Ti_3_C_2_T_x_ (In-situ synthesis)	UV	-	MO	50	92%	[55]
BiOI/Ti_3_C_2_	Vis	20 mg	100 mL 10 mg/L RhB	30	99.8%	[31]
TiO_2_/Ti_3_C_2_	UV	60 mg	60 mL 20 mg/L MO	40	99.6	[56]
TiO_2_/Mxene	UV Vis	10 mg	20 mL 60 mg/L MB	60	96.44% 40.29%	[57]
g-C_3_N_4_/TiO_2_/Ti_3_C_2_	Vis	50 mg	50 mL 20 mg/L MO	120	90.13%	[58]
Defect-Rich Ti_3_C_2_/BiOIO_3_	Vis	10 mg	100 mL (10 mg/L) MO	40	97.6%	[59]
N-Ti_3_C_2_	UV	50 mg	50 mL 20 mg/L MO	40	99.93%	This work

## Data Availability

Not applicable.

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
