# Peer review of "Improving the Photocatalytic Activity of Ti3C2 MXene by Surface Modification of N Doped"

_materials, 2023, doi:10.3390/ma16072836_

Round 1

Reviewer 1 Report

The paper was well-studied and documented.

Author Response

Thank you for your affirmation and support of our work. Best wishes!

Reviewer 2 Report

I think this article should be revised in terms of some technical detail. I will give in more detail below.

-       In the article it is necessary to show in the form of a diagram the entire process of synthesis of N-Ti3C2 catalyst

-       Develop a table to show a comparison of the current study with others.

-       The literature review is inadequate and should be revised. Therefore, the introduction should be expanded and improved. You may find the following:

https://doi.org/10.3390/nano10091734

https://doi.org/10.1016/S1872-2067(20)63630-0

https://doi.org/10.1016/j.apsusc.2021.149176

https://doi.org/10.1016/j.chemosphere.2020.129478

-       Please indicate the limitations of this work. It is also necessary to add some suggestions for further work to overcome the limitations.

-       The novelty of this study needs to be clarified in the introduction.

-       Although the results are quite high in a relatively short degradation time (20 min), in this study a lamp with a power of 1000W was used. How economical is it to use such a high wattage lamp for practical applications in the future.

-        Please specify the wavelength of the UV light emitted under "Photocatalytic performance".

Author Response

   We have revised the manuscript based on the reviewers' comments and have submitted the responses as an attachment.

Reviewer 3 Report

This article needs major revision.

1. Abstract must be more informative. Clearly mention the MO dye in the Abstract.

2. Add the novelty statement in the introduction. Add the dye-related literature in the introduction

3. what was the Ti3C2 yield after HF treatment?

4. 2.4 Photocatalytic performance: write the experimental details clearly, live catalyst mass, conc, volume etc.  

5. Add the Ti3AlC2 XRD. Compare the plane values of Ti3C2 and N-Ti3C2 (1:10) because the 2 thetas have changed a lot.

6. What are the conditions for the scavenger study? If the catalyst degrades the dye, it can also degrade the scavenger molecules, which are also organic species.

7. Authors did not write anything band gap energy of the N-doped materials, and authors must include the band gap anlysis results.

8. The mechanism must be described based on band gap analysis results. The mechanism must mention the role of N doping. 

Author Response

(The authors gave the same response as above.)

Reviewer 4 Report

Section 2.4. Authors should provide more information about the reactor apparatus used in the experiments. Solution volume, material amount, dye solution concentration, reactor type, reactor environment, squeme/real image, etc.

In particular, authors should mention and discuss the dye concentration that was used with respect to the results that were presented. 

A given concentration may offer good results under experimental conditions, while other concentrations may not (mainly in the 10-6 to 10-3 molar range). This should be shown/discussed in detail and a second analysis may require additional experiments to demonstrate limits of functionality.

Is the dye anionic or cationic? How does this interfere with photocatalysis results? Would the results be the same for dyes of opposite ionic nature?

Section 3.2. What are the references for the experimental data of Drx? There is no mention of ICDD cards either in position or in peak intent to substantiate materials that have been prepared and are being experimented. What are the detailed crystallographic informations?

In "the diffraction peaks of the (002) planes corresponding to N-Ti3C2 1:2, N-Ti3C2 1:5, N-Ti3C2 1:8, and N-Ti3C2 1:10 are shifted to 6.94°, 6.8°, 6.79°, and 6.75°, "

How did the authors arrive at this conclusion? There is no explicit basis for this. In addition, the excerpt is poorly written, beginning of sentence with a small letter, etc., and needs to be improved.

In: "This shows that nitrogen was doped during the process."

The result to which the authors refer does not categorically show what is said. This may be a first indication that this is happening. Other experiments like Raman, UV diffuse reflectance, and others need to be performed to demonstrate the defended hypothesis. The XPS spectra data gives some surface interaction information, but does not complement what was previously discussed in the XRD discussion.

Section 3.5 Photocatalytic degradation organic pollutants

Improve the title writing, particularly relating and mentioning what was presented as results of this section.

Fig. 6. The colors relative to each sample are unified in Figs. (a), (b), (c) and (d)? If not, please unify so that graph analysis is facilitated. For example, (b) and (c) complement each other.

What comparisons can be made between the degradation results obtained with studies in the literature, using these materials?

What procedures were performed so that these nanoparticles could be reused? From one cycle to another, what procedures were performed to maintain the same initial conditions?

What ensures that the initial mass remains in the fourth cycle? This is a major problem when using nanoparticles in photocatalysis of organics in solution.

Once 4 cycles are guaranteed without significant loss of efficiency, the authors must obligatorily perform new cycles of photocatalysis until the process efficiency is saturated. The authors must show the efficiency limitations, given the excellent response in the first 4 cases (different concentrations are required, as mencioned early). The authors must explain the facts regarding the loss of efficiency.

Section 3.6

In the proposed mechanism, it is not clear enough which are the positions of the valence and conduction bands of the materials in the eV scale and the respective bandgap and a description of the direction of the transfer of electrons/holes. Is there the emergence of acceptor level, donnor level in doping? How does N doping settle into the host structure? What are the bandgaps of the tested materials?

Author Response

(The authors gave the same response as above.)

Round 2

Reviewer 3 Report

Authors must check the typological errors.

Author Response

Thank you very much for your time and effort in reviewing the manuscript. Based on your suggestions, we have made changes to the typological errors that appeared in the manuscript. Good luck!

Reviewer 4 Report

About the authors response:

"Since there is only a slight efficiency loss after four cycles, if a new photocatalytic cycle is mandatory until the efficiency saturation is reached, it will take a dozen or even dozens of cycles to reach the efficiency saturation, and the cycling experiment is very complicated and time-consuming, and the cycling process is accompanied by mass loss, so we do not cycle the material to the efficiency saturation at present. In addition, cycling experiments are the next step for us to continue our work, so we ask for the reviewers' understanding."

PLEASE INCLUDE THIS LIMITATION AND PERSPECTIVES IN THE CONCLUSION.

Author Response

Thank you very much for your advice. Best wishes!
